# Detecting Parkinson's Disease using Vocal Biomarkers based on Speech Foundation Models

Raymond Brueckner*, Namhee Kwon*, Vinod Subramanian*, Nate Blaylock*, Henry O'Connell*,
Luis A. Sierra†, Simon Laganiere†‡, Ella Lanzaro†§ and Kara M. Smith¶,
*Canary Speech Inc., Provo, UT, USA
{ray, namhee, vinod, nate, henry}@canaryspeech.com
†Department of Neurology, Beth Israel Deaconess Medical Center, Boston, MA, USA
{lsierra1, slaganie}@bidmc.harvard.edu
‡Harvard Medical School, Boston, MA, USA
§Northeastern University, Boston, MA, USA
ellalanzaro@gmail.com
¶Department of Neurology, Boston Medical Center, Boston, MA, USA
kara.smith@bmc.org

*Abstract*—**Parkinson's disease (PD) is the second most common progressive neuro-degenerative disease that leads to loss of motor control, including speech disorder. To discriminate PD from healthy control individuals, we propose an approach based on vocal biomarkers which are derived from pre-trained speech foundation models (SFM) in combination with efficient, shallow downstream classifiers. To validate our approach, we collected a new US English PD dataset in real-life clinical environments, including clinical diagnosis labels. Specifically, focusing on conversational, unconstrained speech, we compare the performance of a variety of off-the-shelf SFMs in combination with different classifiers. We find that a combination of biomarkers derived from the HuBERT Large ll60k SFM and a Random Forest classifier leads to an unweighted average recall (UAR) of 0.97 and an Area-Under-the-Curve (AUC) of 0.97, which proves the validity of the proposed approach.**

*Index Terms*—**Parkinson's Disease, Vocal Biomarker, Foundation Models, Speech For Health, Health Care**

## I. INTRODUCTION

Parkinson's disease (PD) is a progressive neurodegenerative disorder primarily recognized for its motor symptoms, such as tremors, rigidity, and bradykinesia [1]. According to a recent study [2] approximately 900.000 new cases of PD are diagnosed every year in the United States and currently nearly one million people in the US live with PD, this number being expected to rise to approximately 1.2 million by 2023 [3]. This makes Parkinson's the second most common neuro-degenerative disease after Alzheimer's disease.

Diagnosing Parkinson's disease can be challenging due to its complex and varied symptoms. According to a survey by Parkinson's UK[1], 21% of the respondents reported visiting their general practitioner three or more times before being referred to a specialist. At the same time, up to 89% of people suffering from PD are reported to experience speech disorders, commonly referred to as hypokinetic dysarthria [4].

During the prodromal phase[2] of PD, individuals may show emerging symptoms up to five years before the onset of significant motor impairments [5]. The pervasive nature of speech impairments in PD has thus spurred interest among researchers in utilizing vocal biomarkers, since they offer a non-invasive, cost-effective, and efficient means to assess disease presence and progression.

A multitude of proposals were made, aiming to detect PD via an analysis of speech signals using machine learning models. For instance, an earlier study [6] conducted a multi-lingual investigation on the detection of PD based on unvoiced/voiced segmentation on either isolated words or the Pataka speech test. An automated system to predict the severity (UPDRS-III scale) of PD from speech using conventional frame- and utterance-level features was proposed in [7]. A more recent study suggested the use of an end-to-end one-dimensional convolution neural network (1D-CNN) in combination with a Long-Short Term Memory (LSTM) recurrent network [8] for Pataka, read text, and free speech. Some researchers have also specifically focused on investigating prosodic, articulation, and phonetic information [9], [10] to discriminate between PD and healthy control (HC). Voice source information features were evaluated on a shallow Support Vector Machine (SVM) model and an 1D-CNN end-to-end system in [11].

In recent years speech foundation models (SFM) have seen widespread use in numerous research areas, since they offer generalization across tasks, multi-linguality, improved robustness, and efficiency in low-resource settings, among others. This is due to the fact that they commonly are trained on vast amounts of audio in a self-supervised fashion. One of the first approaches of adopting SFMs in PD detection was proposed by [12], which compared different feature-model combinations on five different languages and reported best F1 scores as high as 85% for SFM features in the mono-lingual

---

[1]https://www.parkinsons.org.uk/news/poll-finds-quarter-people-parkinsons-are-wrongly-diagnosed

[2]Prodromal: the phase where some symptoms are present, but they are insufficient to for a clinician to make a diagnosis.

setup. La Quatra et al. [13] compared a variety of foundation models on the Spanish PC-GITA dataset and obtained highest average Accuracy, F1 score, and AUC values of 0.82 based on the WavLM Base model. Most recently, a comparison between conventional and SFM features was presented in [14], where the features were averaged across the time dimension, followed by a downstream single-layer feed-forward classifier. They reported highest accuracy of 0.85 with Whisper embeddings on the PC-GITA database, when using fine-tuning and low-rank adapters.

One potential shortcoming of these previous approaches is their partial or inhomogenous use of constrained speech, such as specific vocalization tests, read speech, etc, which is inconvenient and impractical in real-life conversational doctor-patient interactions [15]. Also, some approaches involve fine-tuning of the pre-trained SFMs, which can prove difficult in small data regimes. In this light, our contribution presents the following novel aspects:

1) We introduce a new US English PD dataset, collected in a real-life clinical environment and providing labels from clinical neurologists.
2) We present comparative results for a substantial number of off-the-shelf speech foundation model variants, i.e. without applying any additional fine-tuning, in combination with efficient, shallow downstream classifiers. This alleviates the problem of overfitting in small-data regimes.
3) We assess the feature-model combinations on unconstrained, free, and open-vocabulary speech, which is particularly beneficial in screening and monitoring scenarios and unconstrained doctor-patient conversations.

The paper is organized as follows: Section II describes the newly introduced dataset. Section III introduces the methods used in this study. Section IV describes the experiments and results obtained, and Section V summarizes our findings.

## II. Dataset

In the realm of audio-based health care research, datasets predominantly are very small in comparison to other speech applications, such as automatic speech recognition (ASR), speaker recognition, or foundation model pre-training, where audio data in the order of thousands to several hundred thousands of hours of speech is used. The reasons for the scarcity of audio data are manifold and include privacy and ethical concerns, annotation complexities and cost, as well as legal and institutional constraints. Also, the few publicly available datasets for the detection of Parkinson's disease can often only be used for research purposes under very restrictive license terms, preventing the wide-spread development and application of solutions to the public.

For these reasons, we present a new US English Parkinson's disease dataset, collected at the Beth Israel Deaconess Medical Center (Boston, MA, USA) and the University of Massachusetts Chan Medical School (Worcester, MA, USA), where speech audio was recorded at the approved research locations within the institutions. All participants were recruited by the institutions and provided written consent for a one-time study visit. The study was approved by the local Institutional Review Boards. Eligibility for the PD group required a confirmed diagnostic status and an age of 18 or older, while individuals with developmental or acquired dysarthria unrelated to PD were excluded. Healthy Control (HC) participants were required to be at least 18 years old and free from any current or previous PD or other neurodegenerative disorder. Each participant completed a 6-minute tablet-based speech protocol consisting of both constrained and unconstrained tasks. The unconstrained tasks included a descriptive analysis (*Cookie Theft Picture*), a procedural narrative (*"Tell me how to make a sandwich"*), and an open-ended question (*"How are you feeling today?"*). The constrained tasks involved paragraph reading (*The Caterpillar Passage* [16]) and all components of the Stroop Color Word Test [17].

The Caterpillar Passage was chosen over the *My Grandfather Passage* [18] due to its optimized design to evaluate speech deficits in movement disorders, offering advantages such as increased complexity, greater phoneme coverage, more varied utterance lengths, lower reading level requirement, and increased prosodic variation. The Stroop task was included for its sensitivity to early deficits in PD, assessing lexical processing, articulatory speed, and susceptibility to interference.

The participants were seated in a quiet room with a Samsung Galaxy Tab S8 positioned on a stand approximately 12 inches from their mouths, using the tablet's internal microphone for recording. A proprietary application guided participants through the protocol, with a study coordinator instructing them to stay focused on each task until completion. All voice samples were recorded at a sampling rate of 44.1 kHz with 16-bit resolution, and subsequently downsampled to 16 kHz. Each assessment was assigned a label of either *PD* (suffering from Parkinson's Disease) or *HC* (healthy control), representing the outcome of the clinical assessment of the respective participant. Incomplete assessments, including empty voice recordings, were excluded from the analysis. In total the PD dataset contains 162 subjects. Table I shows the label distribution per class.

TABLE I
Distribution statistics for the target categories: given are number of subjects, gender (f: female, m: male), and age in years (mean ± standard deviation).

| Category | Nr. Subjects | Gender (f $\mid$ m) | Age |
|---|---|---|---|
| PD | 81 | 24 $\mid$ 57 | 68.2 ± 8.8 |
| HC | 81 | 43 $\mid$ 38 | 40.8 ± 15.0 |

As alluded to above, in this study we wanted to assess the feasibility of PD detection from speech in a real-life clinical environment based on unconstrained, free, and open-vocabulary speech, since this offers non-invasive, unconstrained doctor-patient conversations without any need for specific syllable, vocalization, or reading tests. For this reason, we decided to solely focus on the open-ended question type *"How are you feeling today?"* in the dataset. This approach

allows for screening scenarios, where audio analysis is performed on natural conversational speech from the patient in the background, without the need for elicitation of specific reading and vocalization tests. Table II shows the audio duration statistics for the two categories *PD* and *HC*.

TABLE II

AUDIO DURATION STATISTICS IN SECONDS FOR THE TARGET CATEGORIES. DENOTED ARE THE MEAN DURATION WITH STANDARD DEVIATION AND MINIMUM AND MAXIMUM DURATION.

| Prompt | Category | Mean $\pm$ Std | [Min, Max] |
|---|---|---|---|
| *How Are You?* | PD | 37.6 $\pm$ 12.0 | [12.2, 66.5] |
| | HC | 39.9 $\pm$ 16.5 | [2.2, 80.4] |

## III. METHODOLOGY

### A. Data Organization

Due to the limited size of our dataset and to prevent sampling bias we followed a stratified k-fold cross-validation (CV) scheme, with $k = 5$, with stratification across category labels (i.e. HC vs. PD) and gender. Each fold consisted of a train, validation, and test split, containing 60%, 20%, and 20% of the data, respectively. Since we used only one audio recording per subject (patient) in our study, each subject appeared either in the train, validation, or test split exclusively.

### B. Audio Pre-processing

In this study we focused on vocal biomarkers representing only voice characteristics, excluding information such as pause lengths, frequency, etc. Non-speech parts of the audio signal might also contain irrelevant or even disturbing elements, such as background noise. Therefore, we applied the Silero voice activity detector (VAD)[3], version 5, with the default parametrization, to each audio signal before further processing.

### C. Features

To allow for a thorough comparison, we generated audio embeddings using four well-known speech foundation models for a number of variants. The raw audio input signals were presented as input to the models and their last hidden layer activations (prior to any further processing, such as tokenization) represent the embeddings, which were used in the assessments described in further sections. In particular, we assessed the embeddings for the following models:

**Wav2Vec2**, proposed in 2020 [19], is a transformer-based framework and one of the earliest (self-supervised) audio-based foundation models. It is trained with a constrastive learning objective where the model distinguishes between correct and incorrect latent representations of audio segments. We experimented with five different per-trained variants of Wav2Vec2 (each of embedding dimensionality $D = 512$) spanning a wide range of number of parameters between 94

M and 2.1 B. The specific variants are (a) *Base*[4], (b) *Base 960h*[5], (c) *Large 960h*[6], (d) *XLS-R 300m*[7], and (e) *XLS-R 2b*[8].

**WavLM** builds upon Wav2Vec2 introducing several improvements to enhance performance across diverse tasks, in particular improved performance in noisy environments [20]. We assesed two pre-trained models, namely (a) the *Base*[9] model, comprising 94 M parameters, and (b) *Large*[10] with 315 M parameters. Both models have an embedding dimensionality of $D = 512$.

**HubERT** (Hidden-unit BERT) is a self-supervised learning model proposed in [21] for general audio processing. It builds upon concepts from BERT (Bidirectional Encoder Representations from Transformers) [22], but is adapted for audio signals instead of text. We evaluated two differently sized flavors of this model, namely the *Base ls960*[11] model featuring 94 M parameters and embedding dimensionality $D = 768$ and the larger *Large ll60k*[12] model with 315 M parameters and $D = 1024$.

The **Trillsson** models were proposed in [23] as small and efficient networks to generate universal paralinguistic speech representations, obtained via knowledge distillation from the previously published much larger CAP12 model [24]. These models have been used in related healthcare domains such as the detection of anxiety and depression [25] or mild cognitive impairment [26]. There exist five variants of the Trillsson model with different architecture types and number of parameters, of which we assessed versions (a) v1[13], the smallest of the Trillsson models, which is a ResNet [27] architecture with 5.5 M parameters; (b) v2[14], a slightly larger network (8.1 M parameters) based on an EffNetv2 [28] architecture; (c) v3[15], also an EffNetv2 architecture with 21.5 M parameters, All three investigated Trillsson models generate embeddings of dimensionality $D = 1024$.

As explained in Section III-D, in this study we focused on shallow machine learning models without any specific temporal modeling. Since all foundation models generate embeddings at a relatively high frame rate, we averaged all embedding vectors across time to obtain a single representation of the input audio signal. The resulting mean embedding will be referred to as the *suprasegmental* feature vector and represents input to the classification models described in the following.

---

[3] https://github.com/snakers4/silero-vad

[4] https://huggingface.co/facebook/wav2vec2-base
[5] https://huggingface.co/facebook/wav2vec2-base-960h
[6] https://huggingface.co/facebook/wav2vec2-large-960h
[7] https://huggingface.co/facebook/wav2vec2-xls-r-300m
[8] https://huggingface.co/facebook/wav2vec2-xls-r-2b
[9] https://huggingface.co/microsoft/wavlm-base
[10] https://huggingface.co/microsoft/wavlm-large
[11] https://huggingface.co/facebook/hubert-base-ls960
[12] https://huggingface.co/facebook/hubert-large-ll60k
[13] https://www.kaggle.com/models/google/trillsson/tensorFlow2/1
[14] https://www.kaggle.com/models/google/trillsson/tensorFlow2/2
[15] https://www.kaggle.com/models/google/trillsson/tensorFlow2/3

BINARY CLASSIFICATION TEST SET RESULTS OF SUPRASEGMENTAL FEATURES DERIVED FROM DIFFERENT ACOUSTIC FOUNDATION MODELS APPLIED TO A RANDOM FOREST MODEL. $D$ DENOTES THE DIMENSIONALITY OF THE SUPRASEGMENTAL FEATURE VECTOR, THE NUMBER OF PARAMETERS ARE IN MILLIONS (M) OR BILLIONS (B), UAR DENOTES THE UNWEIGHTED AVERAGE RECALL, SENS THE SENSITIVITY, SPEC THE SPECIFICITY, AND AUC THE (ROC) AREA-UNDER-THE-CURVE.

| Features | Variant | $D$ | Nr. parameters | UAR | Sens | Spec | AUC |
|---|---|---|---|---|---|---|---|
| Trillsson | v1 | 1024 | 5.0 M | 0.94 | 0.94 | 0.94 | **0.98** |
| | v2 | 1024 | 8.1 M | 0.94 | 0.95 | 0.93 | **0.98** |
| | v3 | 1024 | 21.5 M | 0.94 | 0.96 | 0.91 | 0.97 |
| **HuBERT** | Base ls960 | 768 | 94 M | 0.93 | 0.94 | 0.93 | 0.97 |
| | **Large ll60k** | 1024 | 315 M | **0.97** | **0.98** | **0.96** | 0.97 |
| Wav2Vec2 | Base | 512 | 94 M | 0.86 | 0.86 | 0.86 | 0.94 |
| | Base 960h | 512 | 94 M | 0.86 | 0.86 | 0.86 | 0.94 |
| | Large 960h | 512 | 315 M | 0.87 | 0.88 | 0.86 | 0.94 |
| | XLS-R 300m | 512 | 315 M | 0.88 | 0.86 | 0.89 | 0.93 |
| | XLS-R 2b | 512 | 2.1 B | 0.86 | 0.86 | 0.85 | 0.91 |
| WavLM | Base | 512 | 94 M | 0.80 | 0.73 | 0.88 | 0.86 |
| | Large | 512 | 315 M | 0.87 | 0.85 | 0.89 | 0.93 |

## D. Machine Learning Models

As alluded to in Section II the size of available datasets in the healthcare domains are comparably small. This prevents the use of large supervised models - in contrast to the ubiquitous huge models used in present-day AI - since otherwise one immediately faces the problem of overfitting. One approach to alleviate this situation is the use of models which were pre-trained in an un- or self-supervised fashion on large amounts of audio data - this idea is pursued by the use of foundation models, as described in Section III-C. The second approach is to apply tried-and-tested shallow (or small) downstream models, which perform classification directly on the (suprasegmental) audio embeddings. These models feature relatively few trainable parameters, hence tend to suffer little from overfitting, and were used to assess the effect of the foundation models from Section III-C in the detection and classification of Parkinson's disease.

We built and compared five different, well-known model architectures using the sklearn [29] Python framework, namely Support Vector Machine (SVM), Logistic Regression, Random Forest (RF), XGBoost [30], and a feed-forward deep neural network (FF-DNN). Model and training parameters were tuned on the validation set (cf. Section III-A) and then fixed. For the experiments in Section IV we trained SVMs with a radial basis function (RBF) kernel and a complexity parameter $C = 10$. For the logistic regression model we used the LBFGS optimizer with $L_2$ regularization. The Random Forest classifier was constructed using 100 estimators, the Gini impurity criterion, and the maximum number of features defined as the square root of the feature (embedding) dimension $D$. For the XGBoost classifier we found the sklearn default parameter settings to be optimal. Finally, we built a 2-layer FF-DNN classifier model using the Keras framework [31] with 320 nodes in the first layer and 160 nodes in the second layer, followed by softmax classification layer.

Where applicable, we used an inverse time decay learning rate schedule with an initial learning rate $lr = 10e^{-3}$, and a decay rate $r_d = 0.1$ for all models.

## E. Measures

Our evaluation metrics include (a) *Sensitivity*, defined as the positive class (PD) recall, (b) *Specificity*, defined as the negative class (HC) recall, (c) the *Unweighted Average Recall (UAR)*, just the average of Sensitivity and Specificity, and (d) the *Area-Under-the-Curve (AUC)*. Note that UAR and standard Accuracy are nearly identical in our study, since our dataset is balanced w.r.t. the labels (cf. Table I).

## IV. EXPERIMENTS AND EVALUATION

### A. Foundation Model Variants

As outlined in Section III we applied a VAD to each audio file to remove non-speech segments and we extracted frame-level audio embeddings via the foundation model under consideration. We subsequently averaged the embeddings to obtain a suprasegmental feature vector as a single representation of the audio input. This feature vector served then as the input to the the chosen binary classification downstream model, which predicted either Healthy Control (HC) or Parkinson's Disease (PD) as the outcome. Since we ran each experiment in a 5-fold CV fashion all predictions were aggregated and the measures were then computed. The models were trained on the train split and model selection and early stopping determined using the validation split. The best model determined in that way was then evaluated on the test split.

Table III depicts the classification results for all the foundation model variants on the Random Forest model. First, one can observe that the Trillsson and HuBERT models outperform the Wav2Vec2 and WavLM models by a large margin, with the latter showing lowest classification performance. Second, Trillsson features perform almost equally well, independent of which variant is used. Although they offer a much lower parameter count (due to distillation), they perform slightly better than the HuBERT Base model, for almost all measures, and particularly Trillsson v1 possesses the often desirable property

of a balanced Sensitivity/Specificity ratio, i.e. $Sens == Spec$. Yet, the best candidate is the HuBERT Large ll60k model, resulting in an impressive binary classification UAR of 0.97, which represents an absolute margin of 3% over the second-best foundation model. It also has the property of a balanced sensitivity-to-specificity ratio.

### B. Normalization of Embeddings

Most audio foundation models have some form of acoustic normalization pre-processing step built into the model to account for differences in loudness and other acoustic variabilities. To assess the invariance of the model embeddings we applied z-score normalization, i.e. zero-mean and unit-variance normalization, to each of the embeddings and retrained the respective Random Forest models. For all of the feature-model combinations shown in Table III the absolute difference between classification performance of the unnormalized and normalized embeddings were $\leq 0.01$ for all measures (UAR, Sensitivity, Specificity, and AUC). This confirms the hypothesis that normalizing the foundation model embeddings does not have any impact on classification performance. This is an important finding, since it shows that the use of acoustic foundation models is robust w.r.t. to conventional, hand-engineered, e.g. prosodic or spectral, features, which often require careful normalization techniques.

### C. Models

Eventually, we used the Hubert Large ll60k foundation model embeddings, which performed best on the Random Forest model, and trained all other model architectures described in Section III-D on them. The respective performance measure comparison is given in Table IV.

TABLE IV
COMPARISON OF TEST SET RESULTS WITH SUPRASEGMENTAL HUBERT LARGE LL60K FEATURES APPLIED TO DIFFERENT MACHINE LEARNING MODELS.

| Model | UAR | Sens | Spec | AUC |
|---|---|---|---|---|
| SVM | 0.95 | 0.95 | **0.96** | 0.95 |
| Logistic Regression | 0.92 | 0.93 | 0.91 | **0.97** |
| **Random Forest** | **0.97** | **0.98** | **0.96** | **0.97** |
| XGBoost | 0.90 | 0.91 | 0.88 | 0.96 |
| Feed-Forward DNN | 0.93 | 0.96 | 0.90 | **0.97** |

The suitability of the Random Forest model is confirmed, as it shows the highest $UAR = 0.97$ of all model types. The best alternative to the RF model is the Support Vector Machine (SVM), but it trails by 2% absolute UAR and also has lower sensitivity. Interestingly, XGBoost, which was proposed as an improvement over Random Forests, performs considerably worse than its counterpart.

## V. CONCLUSION

In this work, we explored vocal biomarkers to detect Parkinson's disease from audio signals, assessing a substantial number of Speech Foundation Model variants and different downstream classification networks.

First, we introduced a new US English Parkinson's disease dataset, collected in a real-life clinical environment and providing clinical labels, focusing on participant's unconstrained, i.e. free and open-vocabulary, speech to allow for non-invasive, unconstrained doctor-patient conversations without any need for specific syllable, vocalization, or reading tests. The main advantage of unconstrained speech is that this type of speech is a natural speaking style of patients in many communication situations between patient and doctors, nurses, and other clinical personnel. Therefore, it can be used in many clinical scenarios and is particularly well suited for screening purposes, where the model constantly listens in the background without specific interaction from neither doctor nor patient. This greatly expands the applicability of audio-based detection of Parkinson's disease. We point out that the age and gender imbalance between the PD and HC groups potentially introduces a bias and we plan to investigate its impact in future work.

We provide comparative results for twelve different off-the-shelf speech foundation model variants in combination with five different shallow downstream classifiers, without any need for additional model fine-tuning.

Our results provide evidence for the viability and effectiveness of leveraging audio-based SFMs as universal embedding extractors in combination with efficient shallow downstream classifiers on the task of Parkinson's detection from speech. With a Random Forest classifier and features derived from the HuBERT Large ll60k SFM we obtain an UAR of 0.97, a balanced ratio of Sensitivity (0.98) and Specificity (0.96), and an AUC of 0.97. Although it is not clear why the large HuBERT models perform considerably better than other SFMs, we conjecture that the reason for the generally high performance of our approach lies in the use of free, unconstrained speech, which might contain more discriminative information about the PD and HC groups. We plan to investigate this aspect further in future research.

We also show that the parameter-efficient Trillsson models achieve high accuracy, with an UAR of 0.94 and an AUC of 0.98, while still featuring a balanced Sensitivity-to-Specificity ratio close to 1. This might be of interest when resource constraints on e.g. edge devices might require small embedding generators.

In future work, we plan to assess our approach using other datasets of Parkinson's disease and compare our results with previously published results on these data. We will also investigate the impact of data imbalance and the bias of demographic factors on the results.

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
