# OpenReview forum: "Detecting Parkinson's Disease using Vocal Biomarkers based on Speech Foundation Models"
_IEEE.org/EMBS/BHI/2025/Conference — BHI 2025_

### Official Review · Reviewer_kwDB · 2025-07-03
**Relevant topic and advanced methodology, but there is a lack of innovation**

**Confidence:** 4
**Clarity Of Writing:** great
**Clinical Significance:** good
**Methodological Novelty:** poor
**Overall Rating:** 3
**Final Rating:** 4

**Experiments And Results:**

fair

**Questions For The Authors:**

Could you clarify if you'd plan to share this dataset publicly?

**Strengths:**

1. Leverages state-of-the-art foundation models from modern machine learning.
2. Addresses a critical and prevalent health condition.
3. Employs a clear and robust evaluation framework for model performance.

**Summary Of The Paper:**

The authors gathered a new dataset of English-speaking patients in real clinical settings, capturing free-form conversations rather than scripted vocal tests. The authors avoided fine-tuning large speech models like previously published papers, and used twelve off-the-shelf speech foundation models directly to extract vocal features and trained five lightweight classifiers on these embeddings. The best performance came from pairing the HuBERT Large ll60k model with a Random Forest, achieving an unweighted average recall of 0.97, sensitivity of 0.98, specificity of 0.96, and an AUC of 0.97. They also demonstrated that smaller, parameter-efficient models like Trillsson can reach similarly high accuracy (UAR 0.94, AUC 0.98), suggesting this approach can work even on resource-limited devices. The study highlights the promise of using free conversation rather than constrained speech tests and sets the stage for future work on why larger models outperform smaller ones in this task.

**Weaknesses:**

1. Although the new dataset is highlighted as a major contribution, the paper doesn’t clarify if or how the data will be shared.
2. The benchmarking is too basic—no comparisons with other leading PD detection methods, making it hard to judge relative effectiveness.
3. Testing traditional multiple classifiers on the same feature set adds little value when their results are nearly identical.
4. There already exist multiple other papers on speech foundation model used for PD diagnosis, and I fail to see any significant difference from those papers to this one. Even though the authors mentioned that other papers: "their partial or inhomogenous use of constrained speech... some approaches involve finetuning of the pre-trained SFMs, which can prove difficult in small data regimes", I can't really see how the unique choices made in this study are significantly more novel compared to existing research.

Minor complaints:
5. The manuscript lacks any diagrams of the data-collection or modeling workflows, and there are no visualizations of results.
6. There’s a LaTeX formatting mistake: the left double quotation mark is used incorrectly.

---

### Official Review · Reviewer_mtRJ · 2025-07-10
**Detecting Parkinson's Disease using Vocal Biomarkers based on Speech Foundation Models**

**Confidence:** 3
**Clarity Of Writing:** great
**Clinical Significance:** great
**Methodological Novelty:** good
**Overall Rating:** 7
**Final Rating:** 7

**Experiments And Results:**

excellent

**Questions For The Authors:**

-

**Strengths:**

The manuscript is well-written, and the approach and evaluations are easy to follow. The performed evaluations are comprehensive and the results are presented in a clear manner.

**Summary Of The Paper:**

The study at hand investigates the detection of Parkinson's disease based on vocal biomarkers. The authors perform a comprehensive study including the generation of a novel dataset, various speech foundation models, and multiple classifiers. The obtained results are very promosing underlining the validity of the approach.

**Weaknesses:**

There are a few typos and sentences that do not make sense in the text, e.g., in the abstract "we compare the performance of a variety of off-the-shelf SFMs combined with different classifiers and find that a combination of biomarkers derived from the HuBERT Large ll60k SFM and a Random Forest classifier we obtain an unweighted average recall (UAR) of 0.97, a balanced ratio of Sensitivity and Specificity, and an Area-Under-the-Curve (AUC) of 0.97."

---

### Official Review · Reviewer_HkWy · 2025-07-14
**Results may be confounded by participant demographics**

**Confidence:** 3
**Clarity Of Writing:** good
**Clinical Significance:** fair
**Methodological Novelty:** fair
**Overall Rating:** 4

**Experiments And Results:**

fair

**Questions For The Authors:**

- Could the results be explained by the fact that the mean age of the control group is ~30 years younger than the patients?
- Could the results be explained by the fact that there are more females in the control group than in the patient group?
- How are the results comparable to studies employing other methods?
- How much more convenient/ how much time is saved by using unconstrained speech rather than standardised constrained speech considering the latter forms part of routine clinical examination anyway?
- Can you explain why you chose to present only the Random Forest results in Table III?

**Strengths:**

- Novel dataset with ground truth provided by clinicians
- Balanced number of controls and patients
- Multiple speech elicitation tasks used to create dataset which is useful for future work
- No leakage of participant in train, validation and test sets
- Comprehensive selection of foundation models and their variants
- Thorough presentation of methods

**Summary Of The Paper:**

- This study wanted to assess the feasibility of PD detection from speech in a real-life clinical environment based on unconstrained, free, and open vocabulary speech, since this offers non-invasive, unconstrained doctor-patient conversations without any need for specific syllable, vocalization, or reading tests.

- The paper:
1. introduced a new US English PD dataset
2. presented comparative results for a substantial number of off-the-shelf speech foundation model variants, i. e. without applying any additional fine-tuning, in combination with efficient, shallow downstream classifiers.
3. presented findings from assessment of the feature-model combinations on unconstrained, free, and open-vocabulary speech

**Weaknesses:**

- In the related work part of the introduction, the paper mentioned multitude proposals made aiming to detect PD via analysis of speech signals, then lists different methods. It would be useful to have a critique of these methods and summary of the research gap at the end of the paragraph to clarify the motivation behind the work.

- Participant metadata collected were only age and gender. It would be useful to collect/ report data on highest qualification attained, occupation, smoking history, and BMI, as these can confound results

- No statistical tests performed to check for evidence of bias in dataset due to age and gender. The HC group mean age is 27.4 years younger and they have more females than the patient group.

- No discussion of strength, limitation and clinical impact of study in Conclusion

- No baseline model results given

---

### Official Review · Reviewer_12Yg · 2025-07-15
**Review of "Detecting Parkinson's Disease using Vocal Biomarkers based on Speech Foundation Models"**

**Confidence:** 3
**Clarity Of Writing:** great
**Clinical Significance:** good
**Methodological Novelty:** good
**Overall Rating:** 5

**Experiments And Results:**

great

**Questions For The Authors:**

The age gap between PD and HC groups is large. Did you assess how much this variable alone could explain the classification performance?

Do you have plans to validate your findings on any external or public datasets to test whether the results generalize beyond your clinical sample?

Typos and writing suggestions:
Opening quotes on page 2 & 3: ” -> ``

**Strengths:**

The experimental design is thorough. Comparisons made across multiple speech foundation models and downstream classifiers.

The paper provides detailed descriptions of the experimental setup, parameters, and evaluation metrics, making the work reproducible if dataset is made available.

The results show clear and meaningful patterns, especially in Table 3, where embeddings are compared. Trillsson and HuBERT embeddings outperform the other models, providing strong evidence for the conclusions. The impact of the choice of classifier is also explored, though differences there are less pronounced.

**Summary Of The Paper:**

This paper studies whether vocal features extracted from off-the-shelf speech foundation models (SFMs) can effectively detect Parkinson’s disease (PD) from open-vocabulary conversational speech. The paper contributes a new clinical speech dataset that includes recordings of both structured and unconstrained speech tasks performed by individuals with PD and healthy controls. Using this dataset, experiments are conducted to systematically compare the performance of averaged embeddings from multiple SFMs combined with simple downstream classifiers. Their experiments show that using HuBERT Large ll60k embeddings with a Random Forest classifier yields the best performance, suggesting a direction for future research in speech-based PD detection.

**Weaknesses:**

The paper lacks external validation or replication on independent datasets. All experiments rely on a single dataset. It is unclear whether the results would generalize to other datasets or recording conditions.